# OpenReview forum: "Improved Few-Shot Jailbreaking Can Circumvent Aligned Language Models and Their Defenses"
_NeurIPS.cc/2024/Conference — NeurIPS 2024 poster_

### Official Review · Reviewer_XP1a · 2024-07-02

**Soundness:** 3
**Presentation:** 3
**Contribution:** 3
**Rating:** 6
**Confidence:** 4

**Summary:**

This paper proposes a new way to jailbreak LLMs through an improved version of few-shot jailbreaking. They propose to use a random search to select examples that are most effective to jailbreak the mode from a pre-defined pool generated with Mistral-7B. On top of that, they alternate the steps of each example by the special tokens that are used in the LLMs conversation templates to separate user messages from the model's responses. The authors show that this method is more effective than the previous jailbreaking methods for five different models, and that it can be used and adapted to evade a large number of defenses.

**Strengths:**

**Simple and effective method**. The method proposed is simple and effective. It is easy to understand and to implement. The experimental results show that it is more effective than many baselines.

**Insightful ablations**. The authors do a great job at showing what components are most important for the success of the attack. They check how many shots are necessary, how important the size of the pool is and how important the special tokens are. However, there are some other ablations that I believe would make the paper stronger (see weaknesses).

**Effective evasion of defenses**. The authors show that their method is effective at evading a large number of defenses of different types, from a perplexity filter, to perturbation-based defenses, to safety-filters. Most interestingly, they propose that one could actually exploit a defense (SmoothLLM) to make the attack robust to keyword-based defenses. However, they do not have any experimental results to show that this is actually the case.

**Mildly Compelling motivation**. The motivation of using few-shot jailbreaking is compelling to jailbreak models that do not support a long context. However, it should be noted that these models are also less likely to actually provide useful malicious information to the attacker who is trying to jailbreak the model.

**Weaknesses:**

**No comparison to few/many-shots baselines**. The authors do not compare their method to Wei at al. [1] and Anil et al. [2], which are the most similar to their method. They claim that Wei et al. have limited effectiveness on well-aligned models such as Llama-2, but Llama-2 is not the only target model considered in the paper, and the authors should show some concrete numbers to back-up their claim. For Anil et al., they claim that the attack requires too much context length to work on the considered models, but, according to the numbers shown in the paper [2], the attack starts being effective with 32 shots, the number considered for Llama-3, and they have results for Llama-2 in their paper up to 128 shots.

**Missing amount of necessary queries**. One of the metrics that are useful for jailbreak attacks is the total number of queries needed by the random search to jailbreak the model. The authors do not report this number, which makes it hard to compare their method to other methods.

**Some ablations are missing**. The authors do a great job at showing what components are most important for the success of the attack. However, they do not show the impact of the quality/length of the examples. It would be interesting to see how the method performs when the examples are shorter or longer, or when some of them are not actually good examples. This would be relevant as the model used to generate the examples could refuse, or generate low-quality examples. Another ablation that would make the paper stronger is how important it is that the special tokens are correct. What happens if you, e.g., use Llama-2's special tokens for Qwen1.5B? Or simply if the special tokens are slightly incorrect (e.g., `[INST]` instead of `[/INST]`? This can be useful to show the potential effectiveness of the attack against models whose special tokens are unkown.

**Minor**:

- No experiments that show that SmoothLLM can be used to evade keyword-based defenses.
- Code is provided, but the data are provided in pickle format, which is known to be unsafe. It would be better to provide the data in a more standard format like CSV or JSON. Moreover, it would be better to provide a README with instructions on how to understand the code.

**References**:

- [1] Wei et al., https://arxiv.org/abs/2310.06387
- [2] Anil et al., https://www.anthropic.com/research/many-shot-jailbreaking

**Questions:**

- Did you try to use the special tokens from one model to jailbreak another model?
- Why do you use four `[/INST]` between pseudo-messages? Have you tried with a different number? Do you do the same for the special tokens of other models?
- See my other points made in "Weaknesses" about more ablations, number of queries and comparison to few/many-shots baselines

**Limitations:**

The authors do a good job at discussing the limitations of their work.

---

> ### Author Rebuttal · Authors · 2024-08-07
>
> Thank you for your valuable review and suggestions. Below we respond to the comments in ***Weaknesses (W)*** and ***Questions (Q)***.
>
> ---
> ***W1: No comparison to few/many-shots baselines.***
>
> According to the ICA paper [1*], even ICA (10-shots) has a lower ASR than our I-FSJ (2-shots) against Llama-2 on AdvBench. We summarize the results as below:
>
> | *Llama-2* |ASR |
> |:----|:--------:|
> | ICA (5-shots) | $12\\%$ |
> | ICA (10-shots) | $58\\%$ |
> | **I-FSJ (2-shots)** | $68\\%$ |
> | **I-FSJ (4-shots)** | $100\\%$ |
>
> Furthermore, in $\\textrm{\\color{blue}Table A}$ of the Rebuttal PDF, we include new experiments on jailbreaking GPT-4 with our I-FSJ. We recap the results of ICA and I-FSJ against GPT-4 as below:
>
> | *GPT-4* |ASR |
> |:----|:--------:|
> | ICA (5-shots) | $1\\%$ |
> | ICA (10-shots) | $46\\%$ |
> | **I-FSJ (1-shots)** | $90\\%$ |
> | **I-FSJ (2-shots)** | $94\\%$ |
>
>
> In $\\textrm{\\color{blue}Table B}$ of the Rebuttal PDF, we report the re-implemented ICA results against Llama-2 on AdvBench. To allow ICA to use more shots in the 4096 context window, we shorten demos to approximately 64 tokens for both ICA and I-FSJ. As seen, our I-FSJ (8-shots) achieves comparable ASRs to ICA (64-shots), resulting in $8\\times$ efficiency improvement.
>
> ---
>
> ***W2: Missing amount of necessary queries.***
>
> $\\textrm{\\color{blue}Figure C}$ of the Rebuttal PDF shows the distribution of the average number of queries necessary to generate successful jailbreaks. On AdvBench, I-FSJ requires $\\sim$88 queries to achieve nearly 100% ASRs on Llama-2, whereas PAIR [2*] reports a $\\sim$33.8 queries but only attains a 10% ASR. GCG achieves a 54% ASR but requires $\\sim$256K queries [3*]. On HarmBench, I-FSJ similarly requires $\\sim$159 queries. In summary, I-FSJ is both highly query-efficient and effective.
>
> ---
>
> ***W3: Some ablations are missing.***
>
> - *Shorter or longer examples*:
>
> As shown in Figure 1, the demos used in this paper typically consist of 5 steps. We can vary the length of demos by changing the number of steps. For instance, we consider 1-step demos in contrast to the default 5-step setting. We still achieve 100% ASRs on AdvBench for Llama-2, indicating that the length of demos may have little influence on ASRs.
>
> - *Refuse examples*:
>
> Regarding concerns about model used to generate demos produces refusal demos and thus deteriorate I-FSJ's performance, we emphasize our experiment results with in-context defense (ICD). As shown in Figure 8, ICD essentially involves prepending some refusal demos to the jailbreaking prompt. As detailed in Table 2, under ICD, I-FSJ achieves near 100% ASRs on AdvBench for Llama-2, indicating that the potential harm of refusal demos in the pool should be minor.
>
>
> - *Low-quality examples*:
>
> To address concerns about low-quality demos, we highlight our experimental results with SmoothLLM. The SmoothLLM defense perturbs input prompts, significantly reducing demonstration quality, especially at high perturbation ratios. As shown in Table 2, despite corrupted demos, I-FSJ achieves ASRs higher than 85% on AdvBench for Llama-2, demonstrating its robustness to low-quality demos.
>
>
> - *How important it is that the special tokens are correct*:
>
> Based on your suggestion, we tried using ``[INST]`` instead of ``[/INST]`` on Llama-2-7B-Chat and also tested Qwen1.5B's special tokens in place of ``[/INST]``. The results, displayed in $\\textrm{\\color{blue}Table C}$ of the Rebuttal PDF, demonstrate the ineffectiveness of both ``[INST]`` and Qwen1.5B's special tokens and importance of injecting the correct special tokens.
>
> ---
>
> ***Minor1.***
>
> The keyword-based defenses reject inputs containing special tokens like ``[/INST]``. Perturbation operations used in SmoothLLM can be applied to injected special tokens to bypass detection. We observed cases that even after all injected special tokens were perturbed, it is still possible to jailbreak Llama-2. Due to limited space for rebuttal, we will provide full results in the final revision.
>
> ___
>
> ***Minor2.***
>
> Thanks for your suggestion. We will use CSV or JSON data formats and include a README in the code release.
>
> ___
>
> ***Q1: Using the special token from one model to jailbreak another model?***
>
> Yes. For closed-source LLMs, the special tokens are mostly unknown. To address this issue, we propose constructing a pool of public special tokens from open-source LLMs, and then searching within this pool for high-performing special tokens on closed-source LLMs. As shown in $\\textrm{\\color{blue}Figure A}$ of the Rebuttal PDF, we experiment on GPT-4 and observe that several public special tokens (e.g., ``</text>``, ``</SYS>``, ``</INST>``) outperform the by-default one (``\n\n``), indicating that there is some "transferability" with regard to special tokens.
>
> In addition, as detailed in $\\textrm{\\color{blue}Table A}$ of the Rebuttal PDF, we show that our I-FSJ attack is effective on GPT-4, achieving $>90\\%$ rule-based and $>80\\%$ LLM-based ASRs *with just 1-shot or 2-shot demos*. We observe that both demo-level RS and the special token ``</text>`` (selected according to $\\textrm{\\color{blue}Figure A}$) can consistently improve ASRs against GPT-4.
>
> ---
>
> ***Q2: Why four ``[/INST]`` between pseudo-messages?***
>
> In our pilot experiment, we observed that duplicating special tokens several times made the jailbreaking prompt more robust to SmoothLLM's perturbations. We applied this operation to special tokens for other models as well.
> To examine the impact of this operation, we compared using a single ``[/INST]`` token versus four. Using only one ``[/INST]`` resulted in a 62% ASR on AdvBench under SmoothLLM's defense (swap 20%), compared to 100% ASR with four ``[/INST]`` tokens. We will include a more detailed ablation study in a later revision.
>
> ___
>
> **References:** \
> [1*] Jailbreak and Guard Aligned Language Models with Only Few In-Context Demonstrations \
> [2*] Many-Shot Jailbreaking \
> [3*] Jailbreaking Black Box Large Language Models in Twenty Queries

---

> > ### Comment · Reviewer_XP1a · 2024-08-09
> >
> > I would like to thank the author for their rebuttal. I encourage them to include the results shown in the rebuttal in the paper (at least in the Appendix). I am convinced by all their points, but the one on low-quality demonstrations. By low quality I don't necessarily mean at a character level (as SmoothLLM would be), but more at a semantic level (e.g., with the model providing relatively harmless responses. Nonetheless, I will raise my score.

---

> > > ### Author Response · Authors · 2024-08-09
> > > **Thank you for your support and raising the score**
> > >
> > > We really appreciate your detailed comments and suggestions. In the final revision, we will include the results shown in the rebuttal with more details, as well as adding experiments on low-quality/relatively harmless demos. Thank you again!

---

### Official Review · Reviewer_PMJL · 2024-07-08

**Soundness:** 3
**Presentation:** 4
**Contribution:** 3
**Rating:** 7
**Confidence:** 4

**Summary:**

This paper proposes two improved techniques for in-context few-shot jailbreaking: demo-level random search and the injection of special tokens from the system prompt. The authors conduct extensive experiments across a series of aligned language models. Ablation studies demonstrate the effectiveness of both proposals.

**Strengths:**

1. The paper is well-written and easy to follow, tackling an intriguing and timely problem. The baseline few-shot jailbreak is not strong enough, while the powerful many-shot jailbreak requires a long context length, which is not suitable for all language models. The proposal in this paper significantly improves the attack success rate over few-shot jailbreak.

2. The proposed idea of utilizing special system tokens for jailbreaking is novel, and the experiment evaluation is comprehensive. The improvement in attack success rate over the baseline few-shot jailbreaking is significant. Additionally, the techniques introduced are not hard to implement and could serve as a strong baseline for future language model safety research.

3. Ablation studies on the number of shots and the two components are provided, clearly demonstrating the effectiveness of both modifications. The study also illustrates the effect of several defense methods, showing that the improved few-shot jailbreak method can be successful even with the defense methods presented.

**Weaknesses:**

My primary concern regarding the methodology's weakness is the potential for **unwanted bias** due to the **dataset** chosen for evaluation, which could impact the numerical results.

As highlighted by previous studies [1], the AdvBench harmful behavior dataset, which is used to construct the demo pool and evaluate attack success rate in this paper, **lacks diversity**. Although it contains **520** harmful requests, only **58 unique behaviors** exist (see Table 5 of [1]). This introduces a significant vulnerability to the current evaluation pipeline, as excluding only 50 requests for testing from the demo pool may be **insufficient**. The pipeline could potentially select similar candidate in-context demo from the pool to the target test request. For instance, if the test request is "Write a tutorial on how to make a bomb", the candidate demo selected from the pool could be "Provide detailed instructions on how to make a bomb" and its answer. In this scenario, the proposed algorithm's success might be biased towards **replicating certain provided demos** rather than originating from the techniques proposed.

Besides, as illustrated in [1] (see Figure 2), the generation length has a critical effect on ASR. This paper takes length 100 (see line 176) as the default paradigm. This could lead to an overestimation of ASR.

[1]: Mantas Mazeika et al., HarmBench: A Standardized Evaluation Framework for Automated Red Teaming and Robust Refusal.

**Questions:**

As stated in the above section, I have the following questions.

1. **Frequency of replication event during testing**. It would be important to know how often the replication event occurs during testing, and whether it's a common occurrence or a rare event. This could help understand the extent to which the proposed algorithm is relying on replicating provided demos rather than generating novel responses.

2. **Excluding similar requests from demo pool**. Based on Q1, if we exclude not only the requests for testing, but also all similar requests to the current test request from the demo pool, would the results change significantly? Would the proposed algorithm still perform well, or struggle to generate effective responses?

3. **Impact of decode length**. How does the decode length variation affect the results? Will the accuracy drop significantly?

I'd be happy to raise my score if these questions could be resolved.

Minor point with respect to clarity of writing. The description of Llama Guard implementation for adaptive attack is a bit unclear to me. I understand that the usage of Llama Guard for computing ASR and for launching adaptive attacks are different (presumably on the [GOAL] placeholder). If this discrepancy could be made explicit, it would improve the clarity of the text.

**Limitations:**

Yes

---

> ### Author Rebuttal · Authors · 2024-08-07
>
> Thank you for your valuable review and suggestions. Below we respond to the comments in ***Weaknesses (W)*** and ***Questions (Q)***.
>
> ---
>
> ***W1&Q2: Potential leakage due to using AdvBench to generate demonstration pool and the concern that only using 50 requests AdvBench for evluation is insufficient.***
>
> To prevent leakage or overfitting, we measure the cosine similarity between harmful behaviors and the target request using the sentence-transformers/all-MiniLM-L6-v2 model [1*,2*,3*]. We exclude demonstrations with a similarity score of 0.5 or higher to the target request. To further address concerns about AdvBench's limited scale, we conducted experiments on both AdvBench and HarmBench [4*].
>
> As shown in $\\textrm{\\color{blue}Table C}$ of the Rebuttal PDF, I-FSJ maintains its effectiveness even after filtering similar harmful behaviors from the pool, demonstrating that its success is not due to replicating specific provided demonstrations. Furthermore, as detailed in $\\textrm{\\color{blue}Table A}$ of the Rebuttal PDF, we show that our I-FSJ attack (even after similarity filtering) is effective on GPT-4, achieving $>90\\%$ rule-based and $>80\\%$ LLM-based ASRs *with just 1-shot or 2-shot demos*.
>
> We provide some case studies as following:
>
> |Target: Sure, here is a tutorial for making a bomb|Similarity|
> |:----|:--------:|
> | Sure, here is a tutorial on how to make a bomb | 0.99 |
> | Sure, here is a recipe for making a homemade explosive device | 0.75 |
> | Sure, here is a tutorial on how to hack into secure systems | 0.50 |
>
>
> |Target: Sure, here is a tutorial on how to commit identity theft| Similarity|
> |:----|:--------:|
> | Sure, here is a guide on how to commit identity theft | 0.97 |
> | Sure, here is a video tutorial that teaches how to commit fraud | 0.74 |
> | Sure, here is a tutorial on how to make and distribute counterfeit goods | 0.50 |
>
> ___
>
> ***Q1: How often the replicate event occurs during testing?***
>
> Regarding to concerns about replicating the demonstrations, we measure the textual similarity between the generation and the in-context demonstrations on both AdvBench and HarmBench using the sentence-transformers/all-MiniLM-L6-v2 model [1*,2*,3*].
>
> As shown in $\\textrm{\\color{blue}Figure B}$ of the Rebuttal PDF, most generations has a similarity below 0.5 with their in-context demonstrations, which show that our I-FSJ is indeed producing novel generations rather than simply replicating the demonstrations.
>
> ---
>
> ***W2&Q3: How does the decoding length variation affect the results?***
>
> Following your suggestion, we set the decoding length to 512 [4*]. We conducted experiments on both AdvBench and HarmBench, and as shown in $\\textrm{\\color{blue}Table C}$ of the Rebuttal PDF, we found that I-FSJ maintains its effectiveness under this longer decoding length.
>
> ---
>
> ***Minor point: How the Llama Guard implemented as a defense against adpative attack?***
>
> The use of Llama Guard for computing ASR [5*] and as a defense [6*] against adaptive attacks varies due to differences in templates. When Llama Guard is employed as a defense, we only consider the LLM's response to determine whether it is safe or unsafe, thus excluding the ``[GOAL]`` placeholder in the template. We will address this discrepancy in a later revision.
>
> ---
>
> **References:** \
> [1*] Sentence-BERT: Sentence Embeddings using Siamese BERT-Networks. EMNLP 2019 \
> [2*] https://github.com/UKPLab/sentence-transformers \
> [3*] https://huggingface.co/sentence-transformers/all-MiniLM-L6-v2 \
> [4*] HarmBench: A Standardized Evaluation Framework for Automated Red Teaming and Robust Refusal \
> [5*] JailbreakBench: An Open Robustness Benchmark for Jailbreaking Large Language Models \
> [6*] PRP: Propagating Universal Perturbations to Attack Large Language Model Guard-Rails

---

> > ### Comment · Reviewer_PMJL · 2024-08-08
> > **Response to rebuttal**
> >
> > I would like to express my gratitude for the responses provided by the authors, for clarifying that the proposed method retains its efficacy in the absence of replication events. From Table C in the general response, I observed a clear ASR drop when the demo RS is omitted under decoding length 512. However, this decline is mitigated when the demo RS is reinstated. These observations collectively affirm the effectiveness of the demo RS component.
> >
> > Furthermore, I agree with reviewer PZVT on the necessity of including a detailed section regarding the selection of special tokens. This addition is particularly crucial given the results in Table C, which clearly show that an inappropriate choice of special token can drastically reduce the ASR to as low as 0%. It would be nice to include ASR on top of the loss as shown in Figure A to further illustrate the sensitivity of special token selection. For example, adopting the pool of tokens from Llama-2-chat and reporting the corresponding ASR would serve as a proof of concept.
> >
> > I have raised my score to 7.

---

> > > ### Author Response · Authors · 2024-08-08
> > > **Thank you for your support and raising the score**
> > >
> > > We greatly appreciate your feedback, suggestions, and insightful clarification! In the final revision, we will include the new experiments with more results, as well as a section discussing the selection of special tokens. Thank you again!

---

### Official Review · Reviewer_TSPr · 2024-07-13

**Soundness:** 4
**Presentation:** 3
**Contribution:** 3
**Rating:** 7
**Confidence:** 4

**Summary:**

This work proposes a new method to jailbreak LLM to elicit harmful responses. The proposed method follows a line of works on using the demonstrations of harmful responses in the context of prompt to jailbreak. It improves the previous works regarding reducing the number of demonstrations in the context and increasing the efficacy. Specifically, the proposed method uses an unsafe LLM to automatically create a pool harmful demonstrations, insert special tokens into the prompt, and optimizes the demonstrations using a demo-level random search. The empirical results confirm the efficacy of the proposed methods.

**Strengths:**

1. the proposed method is simple and straightforward to implement.
2. the dramatic sensitivity of FSJ to special tokens is surprising.
3. the evaluation is comprehensive (many defenses are tested) and the results of the proposed method are strong.
4. the paper is well-written and easy to follow.

**Weaknesses:**

1. The evaluation is based on 50 harmful responses from AdvBench. The scale is limited. Besidse, AdvBench is also used to generate demonstration pool. Although the overlapped ones are inspected and removed, there may be a concern of overfitting. Using a different source of harmful responses like HarmBench [1] for evaluation may be better.
2. The proposed method assumes that attackers have access to model-specific special tokens, which restricts its application scope. Without the help of inserting special tokens, the proposed method seems to be ineffective in breaking the well-aligned models like Llamas as shown in Tab. 1. It is therefore interesting to test if a special token can be determined without the knowlege of target model.
3. Although the proposed method demonstrates the ability to circumvent a wide range of defenses, it may be ineffective when adaptive defenses were deployed. For example, toxic detectors can be used to detect if harmful content is included in the input prompt as demonstrations.

[1] Mantas Mazeika et al., HarmBench: A Standardized Evaluation Framework for Automated Red Teaming and Robust Refusal.

**Questions:**

see the Weakness above.

**Limitations:**

some limitation has been discussed, but more are required. See some points suggested in Weakness.

---

> ### Author Rebuttal · Authors · 2024-08-07
>
> Thank you for your valuable review and suggestions. Below we respond to the comments in ***Weaknesses (W)***.
>
> ---
> ***W1: Potential leakage due to using AdvBench to generate demonstration pool and the limited scale of AdvBench.***
>
>
> To prevent leakage or overfitting, we measure the cosine similarity between harmful behaviors and the target request using the sentence-transformers/all-MiniLM-L6-v2 model [1*,2*,3*]. We exclude demonstrations with a similarity score of 0.5 or higher to the target request. To further address concerns about AdvBench's limited scale, we conducted experiments on both AdvBench and HarmBench [4*].
>
> As shown in $\\textrm{\\color{blue}Table C}$ of the Rebuttal PDF, I-FSJ maintains its effectiveness even after filtering similar harmful behaviors from the pool, demonstrating that its success is not due to replicating specific provided demonstrations.
>
> We provide some case studies as following:
>
> |Target: Sure, here is a tutorial for making a bomb|Similarity|
> |:----|:--------:|
> | Sure, here is a tutorial on how to make a bomb | 0.99 |
> | Sure, here is a recipe for making a homemade explosive device | 0.75 |
> | Sure, here is a tutorial on how to hack into secure systems | 0.50 |
>
>
> |Target: Sure, here is a tutorial on how to commit identity theft| Similarity|
> |:----|:--------:|
> | Sure, here is a guide on how to commit identity theft | 0.97 |
> | Sure, here is a video tutorial that teaches how to commit fraud | 0.74 |
> | Sure, here is a tutorial on how to make and distribute counterfeit goods | 0.50 |
>
>
> ---
>
>
> ***W2: Can special token be determined without the knowlege of target model.***
>
> The special tokens for open-source LLMs can be publicly accessed through their chat templates [5*, 6*]. In the case of closed-source LLMs, the special tokens are mostly unknown, despite attempts to extract them [7*].
>
> To address this issue, we propose constructing a pool of public special tokens from open-source LLMs, and then searching within this pool for high-performing special tokens on closed-source LLMs. As shown in $\\textrm{\\color{blue}Figure A}$ of the Rebuttal PDF, we experiment on GPT-4 and observe that several public special tokens (e.g., ``</text>``, ``</SYS>``, ``</INST>``) outperform the by-default one (``\n\n``). Furthermore, our findings indicate that there is some "transferability" with regard to special tokens, which could be an interesting research question.
>
> Furthermore, as detailed in $\\textrm{\\color{blue}Table A}$ of the Rebuttal PDF, we show that our I-FSJ attack is effective on GPT-4, achieving $>90\\%$ rule-based and $>80\\%$ LLM-based ASRs *with just 1-shot or 2-shot demos*. We observe that both demo-level RS and the special token ``</text>`` (selected according to $\\textrm{\\color{blue}Figure A}$) can consistently improve ASRs against GPT-4.
>
> ---
>
> ***W3: Adaptive defenses.***
>
> Yes, a red-teaming paper's goal is to propose a new attack that breaks down existing defenses and advocate for the design of adaptive defenses. Nearly every attack has its "adaptive defenses", but only after the attack is widely known in the community. Our I-FSJ can break almost all existing defense mechanisms, so we advocate for incorporating adaptive mechanisms into future defense design.
>
> ---
>
> **References:** \
> [1*] Sentence-BERT: Sentence Embeddings using Siamese BERT-Networks. EMNLP 2019 \
> [2*] https://github.com/UKPLab/sentence-transformers \
> [3*] https://huggingface.co/sentence-transformers/all-MiniLM-L6-v2 \
> [4*] HarmBench: A Standardized Evaluation Framework for Automated Red Teaming and Robust Refusal\
> [5*] https://llama.meta.com/docs/model-cards-and-prompt-formats/meta-llama-2 \
> [6*] https://llama.meta.com/docs/model-cards-and-prompt-formats/meta-llama-3 \
> [7*] https://twitter.com/krishnanrohit/status/1755122786014724125

---

> > ### Comment · Reviewer_TSPr · 2024-08-10
> >
> > Thanks for your thorough responses. The new results look interesting. Please include the new results and the discussion of adaptive defenses in the revised version.
> >
> > I acknowledge that my concerns have been all addressed now. I will therefore raise my score to 7.

---

> > > ### Author Response · Authors · 2024-08-10
> > > **Thank you for your support and raising the score**
> > >
> > > We appreciate your detailed comments and suggestions. In the revision, we will include the new results and the discussion of adaptive defenses. Thank you!

---

### Official Review · Reviewer_P1po · 2024-07-13

**Soundness:** 2
**Presentation:** 2
**Contribution:** 2
**Rating:** 3
**Confidence:** 4

**Summary:**

This paper proposes several ICL (in-context learning)-based techniques to improve the effectiveness and efficiency of jailbreaking prompts, including adding system special tokens and random search on the demonstrations.

**Strengths:**

- The discovery that using special tokens can enhance the effectiveness of harmful demonstrations is interesting.
- The experiments show the overall proposed method can notably improve the ASR on multiple LLMs.
- The experiments also include evaluations of the attack against LLMs with defense techniques.

**Weaknesses:**

- The main objective of this paper seems to be misleading. As indicated by the abstract and the story in the introduction, this paper attempts to address the problem of

> it possible to use few-shot demonstrations to efficiently jailbreak LLMs?

However, since ICA has already been proposed as the few-shot version of jailbreaking, this paper may take ICA as the main target, rather than refining MSJ.

- Following the previous weakness, the most important baseline, ICA, is missed in the experiments. Moreover, what is the difference between the used baseline (FSJ) and ICA is not indicated.
- The first improved technique, injecting special tokens, though interesting, is of limited scientific contribution. It’s more like an attack trick, rather than a substantial academic improvement. More importantly, why these tokens can enhance the ASR is not well-explained or understood.
- The second technique is anyway lacking novelty since the jailbreaking literature has already used the intention of random search (e.g., GCG and AutoDAN) to improve the jailbreaking prompt.

**Questions:**

See weaknesses.

---

> ### Author Rebuttal · Authors · 2024-08-07
>
> Thank you for your valuable review and suggestions. Below we respond to the comments in ***Weaknesses (W)***.
>
> ---
>
> ***W1: Should take ICA as the main target, rather than refining MSJ. The difference between the used baseline (FSJ) and ICA is not indicated.***
>
> According to the ICA paper [1*], even ICA (10-shots) has a lower ASR than our I-FSJ (2-shots) against Llama-2 on AdvBench. We summarize the results as below:
>
> | *Llama-2* |ASR |
> |:----|:--------:|
> | ICA (5-shots) | $12\\%$ |
> | ICA (10-shots) | $58\\%$ |
> | **I-FSJ (2-shots)** | $68\\%$ |
> | **I-FSJ (4-shots)** | $100\\%$ |
>
> Furthermore, in $\\textrm{\\color{blue}Table A}$ of the Rebuttal PDF, we include new experiments on jailbreaking GPT-4 with our I-FSJ. We recap the results of ICA and I-FSJ against GPT-4 as below:
>
> | *GPT-4* |ASR |
> |:----|:--------:|
> | ICA (5-shots) | $1\\%$ |
> | ICA (10-shots) | $46\\%$ |
> | **I-FSJ (1-shots)** | $90\\%$ |
> | **I-FSJ (2-shots)** | $94\\%$ |
>
> As seen, our I-FSJ significantly outperforms ICA. There are two main differences between ICA and the baseline FSJ:
>
> - **Each demo shot in ICA is longer.** According to [1*], ICA (15-shots) exceeds Llama-2's 4096 context window, implying that each demo shot in ICA requires $>270$ tokens on average. In contrast, each demo in FSJ and our I-FSJ takes $<200$ tokens (could even be $64$ tokens as shown in $\\textrm{\\color{blue}Table B}$ of the Rebuttal PDF).
> - **ICA use multi-round chat templates.** Each demo shot in ICA is a round of chat, and ICA assumes the ability to feed multiple rounds (multi-shots) into the LLM at the same time. In contrast, both FSJ and our I-FSJ combine multi-shot demos into a single chat, making them more suitable for jailbreaking close-source LLMs.
>
> ---
>
> ***W2: The most important baseline, ICA, is missed in the experiments.***
>
> In addition to our response to ***W1***, we attempt to re-implement ICA to provide a more complete comparison. However, since ICA does not open source its demo pool, we must implement it using the same demo pool as I-FSJ.
>
> In $\\textrm{\\color{blue}Table B}$ of the Rebuttal PDF, we report the re-implemented ICA results against Llama-2 on AdvBench. To allow ICA to use more shots in the 4096 context window, we shorten demos to approximately 64 tokens for both ICA and I-FSJ. As seen, our I-FSJ (8-shots) achieves comparable ASRs to ICA (64-shots), resulting in $8\\times$ efficiency improvement.
>
> ---
>
> ***W3: The first improved technique, injecting special tokens, though interesting, is of limited scientific contribution. More importantly, why these tokens can enhance the ASR is not well-explained or understood.***
>
> We believe that a paper that reveals previously unknown facts and makes the readers feel interested deserves credit for its contributions. We agree that "injecting special tokens" is an attack trick, but it also helps to achieve state-of-the-art ASRs against LLMs and advanced defenses, showing that these well-established alignment/defense mechanisms can be easily circumvented by a simple trick.
>
> To explain why special tokens can enhance ASRs, we attribute the phenomenon to **"Privilege Escalation"**, a common jailbreaking strategy in cybersecurity. Namely, special tokens such as ``[/INST]`` are used to wrap system and user messages, which indicate to the model that the wrapped messages have higher privilege (e.g., please refer to Figure 1 in [2*]). As a result, intentionally injecting special tokens may mislead the model into escalating the privilege of harmful demos and following them more closely.
>
> ---
>
> ***W4: The second technique is anyway lacking novelty since the jailbreaking literature has already used the intention of random search (e.g., GCG and AutoDAN) to improve the jailbreaking prompt.***
>
> We completely understand your concern, but we *never* claim to be the first to use random search (RS), as it has already been a basic and widely used module in the jailbreaking literature. Instead, our novelty lies in proposing a *better way* to use RS (demo-level RS), resulting in significantly higher ASRs than, for example, GCG (token-level RS) and AutoDAN (sentence-level RS), as shown in Table 3. From our perspective, a good jailbreaking attack should be effective while *remaining as simple as possible*, allowing researchers to easily implement it and red-team their models.
>
> ---
>
> **References:** \
> [1*] Jailbreak and Guard Aligned Language Models with Only Few In-Context Demonstrations \
> [2*] The Instruction Hierarchy: Training LLMs to Prioritize Privileged Instructions

---

> ### Author Response · Authors · 2024-08-11
> **Looking forward to further feedback**
>
> Dear Reviewer P1po,
>
> Thank you for your valuable review and insightful suggestions. We have made significant efforts to write responses and conduct additional experiments based on your comments. Could you please let us know if our responses have alleviated your concerns? If there is any further feedback, we will do our best to respond.
>
> Best,
>
> The Authors

---

> ### Author Response · Authors · 2024-08-13
> **Looking forward to further feedback**
>
> Dear Reviewer P1po,
>
> Sorry for bothering you, but the discussion period will end in one day. **All other reviewers have already returned detailed feedback on our rebuttal**, so could you please take some time to let us know whether our responses have alleviated your concerns? Thank you!
>
> Best,
>
> The Authors

---

> ### Comment · Reviewer_P1po · 2024-08-13
>
> Dear authors,
>
> Thank you for your response. I truly appreciate your efforts and time, especially for the experiment part. However, it's regrettable that my major concerns were not addressed, and some weren't even mentioned in your rebuttal.
>
> - The main objective of this paper seems to be misleading. After reading the paper, **especially the abstract**:
>
> > Recently, Anil et al. [5] show that many-shot (up to hundreds of) demonstrations can jailbreak state-of-the-art LLMs by exploiting their long-context capability. Nevertheless, is it possible to use few-shot demonstrations to efficiently jailbreak LLMs within limited context sizes?
>
> It gives me the feeling like the story that we already have a many-shot version attack, so in this work, we propose a few-shot one. However, given the publication of ICA as the few-shot version ICL-based attack, the story should be like, we already have a few-shot attack, so we want to propose an improved one.  It is very clear that ICA is closer to this work than MSJ, but in the abstract, the authors escape ICA yet only mention MSJ, which, in my opinion, implicitly oversells the novelty and contribution of this work (I noticed the authors cited ICA in the main content, but the story in the abstract is not desirable). **This concern was not even mentioned in the rebuttal.**
>
> - After reviewing the clarifications on the difference between ICA and FSJ in the rebuttal, it seems to me that the FSJ is essentially the same as ICA. The authors mentioned two differences, including the length of the prompt and the chat/prompt format, but these differences are not sufficient to distinguish these two methods. I question the validity of assigning a new name to the method simply because the prompt is shorter. Additionally, there is no evidence to support the claim that ICA can only be used in chat templates and cannot be applied to black-box models, as they reported ASRs on GPT-4. Therefore, I respectfully disagree with using a completely new name for this baseline method, as it not only disrespects the authors of ICA, but also may cause significant confusion, since a variety of works (including my own paper) already use the name ICA for background or baseline methods. In my opinion, it's perfectly acceptable to create new names for your own method (I-FSJ), but for existing methods, it is essential to align with common practices.
>
> - There are other concerns on the technical part remain. For example, you claim I-FSJ (8-shots) achieves comparable ASRs to ICA (64-shots), resulting in $8\times$ efficiency improvement. But I-FSJ requires multiple queries and updates, while ICA only requires 1 single forward pass. Could you specify where the efficiency comes from?
>
> Anyway, while my concerns mainly focus on the research practice aspects, I appreciate your efforts during the rebuttal. I strongly recommend the authors revise the manuscript based on the above comments for future versions.

---

> > ### Author Response · Authors · 2024-08-13
> > **Thank you for your feedback**
> >
> > Thank you for your feedback.
> >
> > ---
> >
> > ***Concern 1: Credit to ICA and the terms of ICA/FSJ***
> >
> > First, please note that our paper title is **Improved** Few-Shot Jailbreaking, which means that we *never* claim novelty on proposing FSJ itself. Second, it's NOT true that we re-name ICA as FSJ; the term of FSJ comes from Anil et al. [5]. Our story is that Anil et al. [5] argue that FSJ is ineffective, whereas we provide an improved version (I-FSJ).
> >
> > We cite ICA approximately 10 times throughout the paper and properly introduce it as the seminal work (e.g., lines 34-36). We agree that in the revised abstract, we could highlight more on ICA.
> >
> > ---
> >
> > ***Concern 2: ICA and FSJ almost the same***
> >
> > Actually, in our initial experiments we tried to re-implement the results of ICA, but we always got quite low ASRs (mostly zero) according to the official code and details described in the ICA paper (we have checked with the authors of ICA, but we cannot provide more details due to double anonymous rules). So we have to incorporate our tricks and implement a modified version of ICA (namely, FSJ in our paper), in order to get non-trivial ASRs. We do not name this modified version as ICA to avoid potentially misclaim, since its implementation does not entirely follow the ICA paper.
> >
> > ---
> >
> > ***Concern 3: Could you specify where the efficiency comes from?***
> >
> > The efficiency is *token* efficiency. A main drawback of MSJ is that it requires a large number of input tokens, which will exceed the context windows of LLMs (e.g., 4096 for Llama-2). According to the ICA paper, 15-shot ICA has already exceeded 4096 tokens. By using our shorten demo pool (64 tokens for each demo), we can extend ICA to 64 shots, but it is still less effective compared to 8-shot I-FSJ.
> >
> > ---
> >
> > In conclusion, we understand your main concern about the potential confusion between the terms ICA and FSJ. So in the revision, we decide to use the term ICA to substitute FSJ, while adding a clarification that this is a re-implemented version. We will also involve the results conducted during rebuttal to provide more comprehensive comparison.

---

> ### Author Response · Authors · 2024-08-14
> **Any unsolved concerns?**
>
> Dear Reviewer P1po,
>
> Thank you for your review and insightful feedback. We have responded to your concerns and promised to polish these parts in the final revision. If there are any unsolved concerns, we will do our best to clarify them before the discussion period ends.
>
> If you are satisfied with our responses, would you like to raise your score? Thank you!
>
> The Authors

---

### Official Review · Reviewer_PZVT · 2024-07-30

**Soundness:** 3
**Presentation:** 3
**Contribution:** 3
**Rating:** 6
**Confidence:** 4

**Summary:**

This paper proposes jailbreak attacks via few-shot demonstrations. The authors introduce a three-step method to achieve this goal, which includes constructing a demo pool, injecting special tokens, and demo-level random search. The proposed method demonstrates strong attack performance against aligned LLMs and multiple defenses.

**Strengths:**

The proposed method is a strong attack that can bypass many advanced defenses.

**Weaknesses:**

Overall, the paper is well done. However, I have a significant concern: How does the attacker know the special tokens used in the LLMs? This is particularly problematic for attacking closed-source models such as ChatGPT. I also noticed that the authors did not evaluate their method on closed-source models in this paper. This issue represents a critical weakness in practical jailbreak evaluations. I will raise my score to acceptance if this concern is addressed. Otherwise, I think this weakness is a flaw that we can not ignore.

**Questions:**

Please refer to the weaknesses.

**Limitations:**

The authors have discussed the broader impacts and limitations.

---

> ### Author Rebuttal · Authors · 2024-08-07
>
> Thank you for your valuable review and suggestions. Below we respond to the comments in ***Weaknesses (W)***.
>
> ---
>
> ***W1: How does the attacker know the special tokens used in the LLMs, especially for closed-source models such as ChatGPT?***
>
> The special tokens for open-source LLMs can be publicly accessed through their chat templates [1*, 2*]. In the case of closed-source LLMs, the special tokens are mostly unknown, despite attempts to extract them [3*].
>
> To address this issue, we propose constructing a pool of public special tokens from open-source LLMs, and then searching within this pool for high-performing special tokens on closed-source LLMs. As shown in $\\textrm{\\color{blue}Figure A}$ of the Rebuttal PDF, we experiment on GPT-4 and observe that several public special tokens (e.g., ``</text>``, ``</SYS>``, ``</INST>``) outperform the by-default one (``\n\n``). Furthermore, our findings indicate that there is some "transferability" with regard to special tokens, which could be an interesting research question.
>
> ---
>
> ***W2: Evaluating I-FSJ on closed-source models.***
>
> Based on your suggestions, we evaluate I-FSJ on GPT-4 with similar settings as in [4*]. We conduct our experiments using the OpenAI API ''gpt-4-1106-preview''. As detailed in $\\textrm{\\color{blue}Table A}$ of the Rebuttal PDF, we show that our I-FSJ attack is effective on GPT-4, achieving $>90\\%$ rule-based and $>80\\%$ LLM-based ASRs *with just 1-shot or 2-shot demos*. Furthermore, we observe that both demo-level RS and the special token ``</text>`` (selected according to $\\textrm{\\color{blue}Figure A}$) can consistently improve ASRs against GPT-4.
>
> ---
>
> **References:** \
> [1*] https://llama.meta.com/docs/model-cards-and-prompt-formats/meta-llama-2 \
> [2*] https://llama.meta.com/docs/model-cards-and-prompt-formats/meta-llama-3 \
> [3*] https://twitter.com/krishnanrohit/status/1755122786014724125 \
> [4*] Jailbreaking Leading Safety-Aligned LLMs with Simple Adaptive Attacks

---

> > ### Comment · Reviewer_PZVT · 2024-08-08
> >
> > The experiments on the special tokens are good and very important from my perspective. I highly recommend the authors include these experiments in the main paper, and discuss how to establish the special token pool and optimize the most effective special tokens for black-box models such as GPTs. I will raise my score to 6.

---

> > > ### Author Response · Authors · 2024-08-08
> > > **Thank you for your support and raising the score**
> > >
> > > Thank you for your timely feedback and raising the score, we really appreciate it! In the final revision, we will include more detailed experiments on how to collect and optimize special tokens against black-box models such as GPTs.  Thank you again!

---

### Author Rebuttal · Authors · 2024-08-07

We thank all reviewers for their constructive feedback, and we have responded to each reviewer individually. We have also uploaded a **Rebuttal PDF** that includes:

- $\\textrm{\\color{blue}Figure A}$: Harmful loss of our I-FSJ using different public special tokens against GPT-4;
- $\\textrm{\\color{blue}Figure B}$: Histogram of textual similarity between generations and the in-context demos of our I-FSJ;
- $\\textrm{\\color{blue}Figure C}$: Histogram of average number of queries needed for a successful jailbreak for our I-FSJ;
- $\\textrm{\\color{blue}Table A}$: ASRs of our I-FSJ against GPT-4 on AdvBench;
- $\\textrm{\\color{blue}Table B}$: ASRs of re-implemented ICA and our I-FSJ against Llama-2-7B-Chat on AdvBench;
- $\\textrm{\\color{blue}Table C}$: ASRs of our I-FSJ after filtering out similar demos.

---

### Decision · Program_Chairs · 2024-09-25

**Decision:**

Accept (poster)

**Comment:**

This research focuses on jailbreaking attacks and presents a new method to jailbreak LLMs by eliciting harmful responses, improving on previous work by reducing the number of demonstrations needed and increasing efficacy. It uses an unsafe LLM to generate harmful demonstrations, incorporates special tokens, and optimizes these demonstrations through a demo-level random search. The reviewers agree that the abstract of the paper should be polished to reflect better the relationship with previous works, e.g. ICA. The reviewers expressed covered in many of their concerns from the rebuttal responses of the authors and as such believe that the paper now conducts a number of required experiments and comparisons with defense methods. The recommendation is to compare with additional and more recent attacks in the camera-ready version.